# A Flexible Platform of Electrochemically Functionalized Carbon Nanotubes for NADH Sensors

**DOI:** 10.3390/s19030518

**Published:** 2019-01-26

**Authors:** Aranzazu Heras, Fabio Vulcano, Jesus Garoz-Ruiz, Nicola Porcelli, Fabio Terzi, Alvaro Colina, Renato Seeber, Chiara Zanardi

**Affiliations:** 1Department of Chemistry, Universidad de Burgos, Pza. Misael Bañuelos s/n, E-09001 Burgos, Spain; jgarozruiz@ubu.es (J.G.-R.); acolina@ubu.es (A.C.); 2Department of Chemical and Geological Sciences, Università di Modena e Reggio Emilia, Via G. Campi 103, 41125 Modena, Italy; fabio.vulcano@unimore.it (F.V.); p.nico96@gmail.com (N.P.); fabio.terzi@unimore.it (F.T.); renato.seeber@unimore.it (R.S.); 3Institute of Organic Synthesis and Photoreactivity (ISOF), National Research Council of Italy (CNR), via P. Gobetti 101, 40129 Bologna, Italy

**Keywords:** single-walled carbon nanotubes, caffeic acid, catechol, NADH oxidation, electrocatalysis, amperometric sensor, voltabsorptometric sensor, spectroelectrochemistry

## Abstract

A flexible electrode system entirely constituted by single-walled carbon nanotubes (SWCNTs) has been proposed as the sensor platform for β-nicotinamide adenine dinucleotide (NADH) detection. The performance of the device, in terms of potential at which the electrochemical process takes place, significantly improves by electrochemical functionalization of the carbon-based material with a molecule possessing an o-hydroquinone residue, namely caffeic acid. Both the processes of SWCNT functionalization and NADH detection have been studied by combining electrochemical and spectroelectrochemical experiments, in order to achieve direct evidence of the electrode modification by the organic residues and to study the electrocatalytic activity of the resulting material in respect to functional groups present at the electrode/solution interface. Electrochemical measurements performed at the fixed potential of +0.30 V let us envision the possible use of the device as an amperometric sensor for NADH detection. Spectroelectrochemistry also demonstrates the effectiveness of the device in acting as a voltabsorptometric sensor for the detection of this same analyte by exploiting this different transduction mechanism, potentially less prone to the possible presence of interfering species.

## 1. Introduction

β-Nicotinamide adenine dinucleotide (NADH) is the co-factor of many enzymes belonging to the class of dehydrogenases [1,2]. For this reason, the development of an amperometric sensor for NADH detection is the first, mandatory step for the analysis of a wide number of chemical species constituting the substrate of NADH-dependent enzymes. Furthermore, abnormal levels of NADH in living cells could indicate altered status of health. As an example, it was observed that NADH concentration in malignant sites of breast tissue is significantly higher than in the non-malignant sites, whereas the opposite occurs for malignant and normal tissues from the oral cavity [3].

Oxidation of NADH at conventional electrodes, namely glassy carbon, Pt, and Au, requires particularly high overpotentials and induces massive passivation of the surface [4,5,6,7]. For this reason, many materials have been developed and studied so far, aiming at realization of efficient amperometric sensors for the detection of this analyte [8,9,10]. Among others, carbon nanomaterials are now attracting great attention; on the one hand, they can act as the catalysts for NADH oxidation [11,12,13,14,15,16,17] and, on the other hand, they can allow the stable anchoring of a proper redox mediator [14,15,18]. In both cases, the occurrence of an electrocatalytic oxidation in charge of NADH leads to considerable advantages, first of all the possibility of obtaining the electrochemical oxidation of this species at low potential values, possibly improving selectivity and sensitivity of the sensor system [19]. In this framework, it is widely accepted that the o-quinone/o-hydroquinone redox couple constitutes an effective redox mediator for NADH oxidation at an anode [8,10,20]. However, stable anchoring on an electrode surface still constitutes an open problem.

Carbon nanotubes are nanomaterials consisting of rolled up graphene nanosheets, possessing the typical tubular structure [14,21,22]. It is usual to distinguish between structures consisting of single graphite sheets, named single-walled carbon nanotubes (SWCNTs), and those characterized by coaxially wrapped nanosheets, named multi-walled carbon nanotubes (MWCNTs). They possess an external diameter ranging between 1 and 2 nm in the case of SWCNTs and up to 100 nm for MWCNTs, and a length in the order of the microscale. One of the most important advantages afforded by their use in the frame of amperometric sensing is the possibility to stable anchor a high number of molecules suitable to induce selective recognition of the target analyte in affinity biosensors or to induce activation of effective redox mediation, in electrocatalytic (bio)sensors [11,12,13,14]. 

In this paper, flexible electrode systems entirely constituted by SWCNTs are proposed as the sensor platform for NADH detection. Similar systems have been previously tested in electrochemical and spectroelectrochemical experiments also devoted to quantitative analyses [23,24,25,26,27]. The crosslinking between SWCNTs present in these devices is at the basis of the good conductivity of the film, necessary to the use as an electrode platform. The performance of the electrode material was improved here by functionalizing SWCNTs with catechol residues. To such a purpose, SWCNTs have been electrochemically oxidized in strong acidic medium [20,28,29] and the resulting SWCNT_ox_ film has been further modified by performing a voltammetric treatment in a caffeic acid (CFA) solution. The effectiveness of the resulting SWCNT_CFA_ surface in activating electrocatalytic processes in charge of NADH oxidation has been ascertained by performing both electrochemical and spectroelectrochemical experiments in absence and in presence of NADH. UV/Vis absorption spectroelectrochemistry has demonstrated to be a very suitable technique to study electrochemical processes occurring at the electrode surface [25,26,27]. These techniques also demonstrate the efficiency of SWCNT_CFA_ in acting as the sensitive element in optical and voltammetric sensors for NADH detection.

## 2. Materials and Methods

### 2.1. Reagents and Materials

CFA (≥98%, Sigma-Aldrich, St. Louis, MO, USA), NADH (98%, Acros Organics, Geel, Belgium), and H_2_SO_4_ (96%, Merck, Darmstadt, Germany) were used as received. Acetate buffer solution (0.1 M, pH = 4) was prepared with acetic acid (100%, VWR, Radnor, PA, USA) and KOH (Panreac, Barcelona, Spain), whereas 0.1 M phosphate buffer solution (PBS, pH = 7) was prepared from sodium dihydrogen phosphate (NaH_2_PO_4_·12H_2_O, VWR, Radnor, PA, USA) and disodium hydrogen phosphate (Na_2_HPO_4_, 99%, Acros Organics, Geel, Belgium). All solutions were prepared daily using ultrapure water, obtained from a Millipore DirectQ purification system provided by Millipore (18.2 MΩ·cm resistivity at 25 °C, Burlington, MA, USA). All reagents were of analytical grade and used as received, without further purification.

SWCNTs (Sigma Aldrich, St. Louis, MO, USA), 1,2-dichloroethane (DCE, 99.8%, Acros Organics, Geel, Belgium), polytetrafluoroethylene membranes (Teflon®, filter pore size 0.45 μm, Millipore Omnipore, Burlington, MA, USA), polyethylene terephthalate film (PET, 175 μm thick, HiFi Industrial Film, Stevenage, UK), silver conductive paint (Electrolube, Leicestershire, UK) for the electrical contacts, and Kapton® tape as insulator of the electrical contacts were used to fabricate the flexible SWCNT electrodes. 

### 2.2. Instrumentation

In-situ time-resolved UV/Vis absorption spectroelectrochemical experiments were performed with a customized SPELEC instrument (Metrohm-DropSens, Llanera, Asturias, Spain) with a halogen/deuterium lamp. DropView SPELEC software (Metrohm-DropSens, Llanera, Asturias, Spain) was used to control the instrument, allowing us to register real-time spectra synchronized with the electrochemical data.

An Autolab PGSTAT30 potentiostat/galvanostat (Ecochemie, Utrecht, the Netherlands) controlled by GPES software has been used to perform amperometric measurements aimed at testing the possible use of the device as an amperometric sensor for NADH detection.

A tip-sonicator (CY-500, Optic Ivymen System, Sabadell, Spain) was used to disperse the SWCNTs in DCE.

### 2.3. Fabrication of SWCNT Flexible Electrodes

SWCNT electrodes were prepared by the filtration-press transfer methodology proposed in previous works [24,25]. Briefly, it consists of vacuum filtering 1 mL of a SWCNT suspension in DCE (5 mgL^−1^), obtained by tip-sonication, through a 0.45 μm pore size Teflon® filter. The obtained SWCNT film was then transferred on a PET support by simply pressing the coating with the fingers. Afterwards, the filter was dried at room temperature and separated from the PET support using tweezers, obtaining a SWCNT film (10 mm diameter) on a flexible PET support. Silver conductive paint was deposited and dried in an oven at 75 °C for 45 min to make the electrical contact. A schematic representation of the electrode construction is reported in Appendix A.

### 2.4. Modification of SWCNT Electrodes

SWCNTs were electrochemically modified by CFA in three consecutive steps, as reported in Appendix A: (1) SWCNT_ox_ was obtained by immersing SWCNT-based electrodes in a 1 M H_2_SO_4_ solution and by performing 10 consecutive voltammetric cycles between −0.50 and +1.70 V at 0.10 Vs^−1^; (2) SWCNT_ox_ was electrochemically functionalized by performing 10 successive voltammetric cycles between −0.10 and +0.90 V at 0.02 Vs^−1^ in a 10^−3^ M CFA, 0.1 M acetate buffer solution (pH = 4); (3) SWCNT_CFA_ was finally obtained by performing 25 voltammetric cycles between −0.10 and +0.60 V at 0.10 V s^−1^ in 0.1 M PBS (pH = 7), in order to remove the CFA molecules weekly adsorbed on the SWCNT surface.

### 2.5. Amperometric Detection of NADH

Amperometric measurements on SWCNT_CFA_-based sensors were performed at a fixed potential of +0.30 V, dipping the electrode in a magnetically stirred solution. The electrochemical cell was completed, in this case, by a Pt wire counter-electrode and an Ag/AgCl/KCl 3 M reference electrode (Amel, Milano, Italy). Aliquots of a 10^−3^ M NADH solution were added to a 0.1 M PBS (pH = 7), in order to vary the concentration between 0 and 4·10^−5^ M. The whole calibration procedure was repeated three times on the same SWCNT_CFA_ electrode in order to evidence possible memory effects and to test the repeatability of the sensor response. Furthermore, three electrodes realized under the same conditions were used to perform calibration aiming at testing the reproducibility of the sensor building up. Repeatability and reproducibility of the sensor response were quantified in terms of relative standard deviation (%RSD).

### 2.6. UV/Vis Absorption Spectroelectrochemical Set-up and Measurement Procedures 

For spectroelectrochemical measurements, the electrode preparation was completed as reported in Appendix A. In particular, two naked 100 μm optical fibers (Ocean Optics) were fixed on the borders of the electrode surface in order to perform spectroelectrochemical measurements in the parallel configuration, as shown in Figure 1 and Appendix A). They were aligned with each other, using Kapton® tape that also simultaneously allowed isolating the electrode’s silver contact and delimiting the geometric area or the electrode. In this cell configuration, the light beam coming from the halogen/deuterium light source passes throughout the first optical fiber (OF1) and arrives at the solution layer (100 μm) just adjacent to the SWCNT electrode surface. The light beam is then directed back to the spectrometer through the second optical fiber (OF2). This instrumental configuration allowed us to register optical variation occurring inside the diffusion layer involved in the electrochemical process, i.e., in close proximity of the electrode surface.

The spectroelectrochemical measurements were performed dropping 50 μL of the sample solution onto the SWCNT electrode, acting as the working electrode (WE) of a three-electrode cell. The sample should cover up the entire SWCNT surface and the two ends of the optical fiber fixed on it. The electrochemical cell was completed by a home-made Ag/AgCl/KCl 3 M reference electrode (RE) and a Pt wire counter electrode (CE).

Cyclic voltammetry (CV) combined with the simultaneous registration of the UV/Vis absorption spectra was chosen as the technique to study the functionalization of SWCNTs and to estimate the performance of the resulting SWCNT_CFA_ electrodes in NADH detection. These spectroelectrochemical experiments were performed in the parallel configuration, as shown in Figure 1, by polarizing the electrode in 0.1 M PBS (pH = 7) in absence and in presence of NADH at concentration values increasing up to 3·10^−4^ M. In particular, CV traces were recorded by scanning the potential between −0.10 and +0.60 V at 0.02 Vs^−1^ potential scan rate, and by concomitantly registering the UV/Vis spectra in the 200–1000 nm range. Solutions at different NADH concentrations were tested by randomizing the order of their analysis, aiming at evidencing possible memory effects, especially due to electrode fouling, or detachment of the chemical functionalization. To test repeatability, three consecutive replicates of the calibration curves were performed using the same SWCNT_CFA_ electrode and the slopes of the three calibration models were compared. The reproducibility was tested using three different CFA modified electrodes, once more comparing the slope of the relevant calibration plots. Also, in this case, repeatability and reproducibility of the sensor response were defined in terms of %RSD.

## 3. Results and Discussions

### 3.1. Spectroelectrochemical Study of Electrochemical Functionalization of SWCNT

The first step toward functionalization consists of the electrochemical oxidation of SWCNTs in a 1 M H_2_SO_4_ solution, to achieve, as suggested by the literature [29], partial breakdown of the graphitic aromatic structure and formation of oxidized residues, namely aldehydic, ketonic, or carboxylic groups, on SWCNTs. These residues are suitable to stably anchor redox-mediating molecules for possible electrocatalysis.

Figure 2a shows the typical CV response registered during the activation step of the SWCNT electrode. At the highest applied potential, a steep and marked increase of the current intensity can be observed, ascribable to the superficial oxidation of the SWCNT structure. Concurrently, increase of the background current occurs scan after scan, as shown in Figure 2b, indicating progressive modification of the electrode surface. As observed, the highest current increase was observed during the first few scans, evidencing that SWCNT oxidation mainly takes place in this phase. From the eighth scan onwards, the increase of the current was significantly lower, suggesting that the oxidation process occurs to a lower and lower extent. 

The second step of this process consists of the electrochemical modification of SWCNT_ox_ by CFA, which occurs by carrying out 10 successive voltammetric cycles in 10^−3^ M CFA solution and 0.1 M acetate buffer (pH = 4). During this step, a Michael addition reaction takes place, consisting of a nucleophilic attack afforded by the carboxylate moieties, generated on the SWCNTs during the activation step, to the double bond in β to the carboxylic group of the CFA [20,30,31,32]. The pH value of the solution has been chosen for the purpose to avoid the competition of other nucleophilic species, such as hydroxyl ions or deprotonated carboxylic groups of CFA, with carboxylic groups of the electrode surface in reacting with CFA molecules. On the other hand, a solution at a lower pH value could induce protonation of the carboxylic groups in the SWCNT surface [31]. CV traces recorded in this step, as shown in Appendix A, were consistent with the presence of an electroactive species that is reversibly oxidized around +0.60 V on the SWCNT_OX_ conducting electrode surface. The UV-Vis spectra concomitantly recorded in the parallel configuration allow us to obtain direct evidence of the species present inside the diffusion layer as a consequence of the electrochemical process, as shown in Appendix A. During the electrode polarization toward more positive potentials, the spectra show the decrease of an absorption band at 350 nm, typical of CFA molecules, indicating consumption of this species at the electrode surface. At the same time, two new bands at 260 and 410 nm, ascribable to the oxidized quinone form, increased in height; they are due to oxidized CFA molecules not anchored to the electrode surface. Due to the reversibility of this electrochemical process, the spectra evolved in the opposite direction in the backward potential scans. 

A subsequent stabilization process was carried out in order to remove CFA weakly linked to the SWCNT_CFA_ electrode surface. Repeated CV scans were performed between −0.10 and +0.60 V at 0.10 Vs^−1^. The occurrence of the cleaning process was testified by voltammetric responses recorded in pure electrolyte solution (0.1 M PBS solution, pH = 7) on the electrode previously modified by CFA, as shown in Figure 3a: the anodic peak current significantly decreased during the first scans, reaching a stable value of about 2·10^−5^ A after ca. 20 scans, as shown in the inset of Figure 3a. Once this stabilization procedure was completed, the SWCNT_CFA_ electrode was suitable to be used for NADH detection. No significant variation of the current peak, in fact, occurred by performing 25 further scans in conditions similar to those reported in Figure 3a, and no improvement in NADH detection was achieved when using SWCNT_CFA_ electrodes obtained after performing this second stabilization procedure.

The spectra recorded simultaneously to CV traces in Figure 3a only show the growth of the absorption band centered at ca. 250 nm, as shown in Figure 3b, due to the oxidized form of CFA. This absorption dramatically increased during the first three scans, owing to desorption of poorly stable CFA molecules from the electrode that diffused into the solution layer adjacent to it, as shown in the inset of Figure 3b. In the following cycles, the amount of CFA species present in the diffusion layer decreased continuously: the diffusion away from the electrode was no more compensated by desorption from the electrode surface. This indicates that the faradaic contribution to the current recorded at the end of this stabilization step was only due to electroactive CFA stably anchored on the electrode surface.

### 3.2. Study of Electrocatalytic Efficiency of Functionalized SWCNT for NADH Oxidation

The effect of SWCNT functionalization with CFA was firstly investigated by performing CV experiments with SWCNT, SWCNT_OX_, and SWCNT_CFA_ electrodes in the pure electrolyte solution, as shown in Figure 4a. As already outlined, electrochemical oxidation of the carbon-based surfaces in a strongly acidic medium induces significant variation of the electrode surface, evidenced by the different values of background current passing from the pristine to the oxidized SWCNTs. Two ill-defined pairs of peaks ascribable to reversible electrochemical processes were observable around +0.10 and +0.20 V in the CV curves on SWCNT_OX_. The latter couple was enhanced by further functionalization by CFA molecules. On the basis of this result, we can reasonably suggest that the process of electrochemical activation of SWCNTs induces breaking of carbon aromatic structure and formation of (poly)phenol residues that can induce reversible redox processes, as in the case of hydroquinone and catechol. Functionalization of the SWCNT surface by CFA molecules was confirmed by the presence of a well-defined anodic/cathodic peak system ascribable to the o-quinone/o-hydroquinone redox couple.

The performance of the three materials in respect to NADH oxidation is compared in Figure 4b. As observed, the use of SWCNT_ox_ led to the remarkable anticipation of the oxidation peak. The peak was quite broad, suggesting that it was relative to two different electrode processes at potential values close to those observed in the voltammogram registered in the pure electrolyte. This conclusion is supported by the electrochemical responses obtained for NADH oxidation at SWCNT_CFA_: strong increase of the current peak at +0.25 V was observed, suggesting that catechol residues introduced by functionalization with CFA were effective in inducing electrocatalytic oxidation of NADH. In conclusion, the electrochemical results suggest that SWCNT_CFA_ is the most suitable material to act as the active element of sensors for NADH determination.

### 3.3. Use of SWCNT_CFA_ as an Amperometric Sensor

On the basis of the previous results, the performance of SWCNT_CFA_ has been tested for the possible use as an amperometric sensor. To this aim, the electrode was polarized at the fixed potential of +0.30 V, i.e., at a potential value at which the electrochemical process was limited by the mass transfer of electroactive species to the electrode surface; the oxidation current was recorded concurrently to subsequent increases of the NADH concentration in solution, as shown in Appendix A. Calibration was replicated three times on the same electrode, using seven concentration values in each regression, in order to test response repeatability, as shown in Figure 5. The sensor showed a linear correlation of the current value vs. NADH concentration, with a slope 5.26·10^−2^ A·M^−1^. An R^2^ value of 0.9985 and a regression standard deviation (S_y/x_) of 2.5·10^−8^ A indicate that the model fits suitably to the experimental data. The accuracy of the calibration model was evaluated by comparing the true values of different NADH solutions and the predicted ones, obtaining a linear model with slope and intercept values not significantly different from 1.0 and 0.0, respectively. This indicates that the calibration model is unbiased.

Furthermore, the %RSD of the slope values of three regressions on the same electrode (with a mean value of 5.35·10^−2^ A·M^−1^) resulted as 8.7% (n = 3). This value denotes repeatability of the experiments, indicating that the same electrode can be used in at least 27 consecutive experiments without significant changes to its surface.

To evaluate the reproducibility of the building up process, three amperometric calibration curves were computed using three different SWCNT_CFA_ electrodes. The mean of the slopes resulted as 5.56·10^−2^ A·M^−1^; a relative standard deviation (RSD) of 9.8% (n = 3) denotes that the overall fabrication process of SWCNT_CFA_ electrodes is reproducible. 

Finally, the NADH concentration in three test samples, the true value being 1.09·10^−5^ M, was estimated. The confidence interval resulted as [1.01 ± 0.30]·10^−5^ M (RSD = 12%, n = 3), indicating high accuracy and precision.

### 3.4. Use of SWCNT_CFA_ as a Voltabsorptometric Sensor

SWCNT_CFA_ electrodes were also tested for a possible use as voltabsorptometric sensors for quantitative determination of NADH. Potential sweep techniques, in fact, are preferred to amperometry at a constant potential when different electroactive species are present in the same matrix. Even better selectivity can be reached by recording, at any applied potential, the relevant absorption spectrum. This approach aims at further improving the selectivity of the sensor system developed here [27].

In order to verify the effectiveness of SWCNT_CFA_ in NADH detection using a potential scan technique, both electrochemical and spectroscopic signals have been concomitantly acquired in NADH solutions at different concentrations. Figure 6 displays the spectroelectrochemical responses obtained during the electrochemical oxidation of 3·10^−4^ M NADH in 0.1 M PBS, scanning the potential between −0.10 and +0.60 V at 0.02 Vs^−1^. The spectra show a band with a maximum at 260 nm, which increased during the oxidation of NADH. It is related to the π→π* transition of the adenine aromatic ring of the electrogenerated NAD^+^. Concomitantly, the absorption band at 340 nm, ascribable to the n→π* transition of the dihydronicotinamide portion of NADH [33], decreased due to the consumption of this species at the electrode surface. The same experiment, when carried out in pure 0.1 M PBS, i.e., in the absence of the analyte, does not show either of these two bands, supporting the explanation given above, as shown in the inset of Figure 6a.

The trend of these two bands is in good agreement with the voltammetric responses, as suggested by the perfect overlap of electrochemical signal with the derivative voltabsorptogram (DVA) at 340 nm and at 260 nm, as shown in Figure 6b and Appendix A, respectively).

When considering linear sweep voltabsorptograms (LSVA) recorded in the different NADH solutions, the correlation between the peak current and the NADH concentration was very poor, as suggested by a correlation coefficient value around 0.75, as shown in Appendix A. These results indicate that the voltammetric technique is not appropriate to determine NADH. This conclusion should be ascribed to the fact that the voltammetric signal actually combines contributions of both faradic and non-faradic processes, being also very difficult to trace a good baseline, and, consequently, to properly estimate the height of the anodic peaks. 

At variance with voltammetric responses, spectroscopic variations in the UV/Vis spectral region were only directly ascribable to NADH consumption in the diffusion layer, as shown in Figure 7a, leading to a better linear relationship between absorbance and NADH concentration. Since DVA traces were in very good agreement with the electrochemical responses due to NADH oxidation, spectroscopic absorption can be also used for the quantitative estimation of this species. Figure 7b shows absorbance variations at 340 nm collected during the LSVA experiments performed in solutions at different NADH concentrations. Similar behavior was found for the band peaking at 260 nm. Since absorbance changes at 340 nm are twice those observed at 260 nm, that wavelength was chosen for more sensitive analyses. 

The calibration procedure was replicated three times with the same electrode, to evaluate the response repeatability, as shown in Figure 8. Solutions at different NADH concentrations were randomly tested, in order to minimize the bias induced by possible memory effects. A good correlation was found with the linear regression model, relating the values of absorbance at 340 nm to the NADH concentration when the potential applied during the voltabsorptometric experiment was +0.60 V. The slope of the calibration model was −251.0 M^−1^ (R^2^ = 0.9986, S_y/x_ = 1.02·10^−3^ M^−1^). The repeatability, estimated in terms of %RSD value, resulted as 9.7% (n = 3), indicating that the same electrode can be used in different calibration procedures.

To test the reproducibility of the activation-modification procedure of the SWCNT electrode surface, the mean slope values of three linear regression models were calculated with three different electrodes. Table 1 displays the figures of merit of these calibration models. Good reproducibility was estimated on the basis of the RSD value of 2.8% (n = 3) relative to the mean of the three slopes (−267.3 M^−1^). 

Finally, the concentration of three samples of NADH with a true concentration of 1.25·10^−4^ M was estimated. The interval of confidence obtained resulted as [1.24 ± 0.19]·10^−4^ M (RSD = 6.1%, n = 3). The mean value found testifies the accuracy, and the RSD value and the low relative error in the determination of the NADH concentration, resulting lower than 1%, denotes high precision of the procedure followed.

Reliability of the calibration model was also verified by repeating the previously described procedure with a further eight NADH solutions. The concentration values predicted by the model were plotted vs. the true values, exhibiting a linear correlation with slope and intercept values not significantly different from 1.0 and 0.0, respectively. This analysis indicates the absence of bias.

## 4. Conclusions

The results reported in this paper demonstrate that SWCNTs electrochemically functionalized by CFA molecules are suitable to act as the sensing element for detection of NADH. In particular, the presence of catechol residues as a consequence of the electrochemical functionalization leads to electrocatalytic oxidation of NADH, inducing significant anticipation of the electrochemical process to particularly low potential values. Furthermore, deposition of SWCNT on a flexible substrate constitutes an interesting added value for the possible use of this device in the frame of sensors, especially when used ‘in the field’. The device can be used both as an amperometric and as a voltabsorptometric sensor, depending on the analytical frame. In particular, the voltabsorptometric sensor can be of great interest when analyzing matrices containing different electroactive species that can be oxidized at similar potential values but absorb radiation in different spectral ranges.

## Figures and Tables

**Figure 1 sensors-19-00518-f001:**
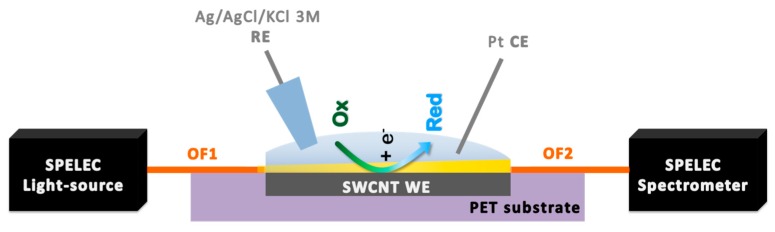
Schematic view of the UV/Vis absorption spectroelectrochemical set-up in parallel configuration. WE: single-walled carbon nanotube (SWCNT) working electrode, CE: Pt counter electrode, RE: Ag/AgCl/KCl 3 M reference electrode, OF1: Naked optical fiber that guides the light beam from the source cell to the solution, OF2: Naked optical fiber that guides the light beam from the solution to the spectrometer, PET: polyethylene terephthalate.

**Figure 2 sensors-19-00518-f002:**
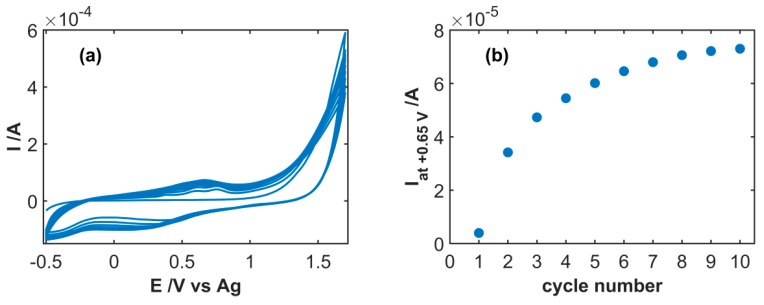
(**a**) Ten consecutive cyclic voltammetry (CV) traces collected during SWCNT oxidation in 1 M H_2_SO_4_ at 0.1 Vs^−1^; (**b**) evolution of the current intensity at +0.65 V in the forward scan.

**Figure 3 sensors-19-00518-f003:**
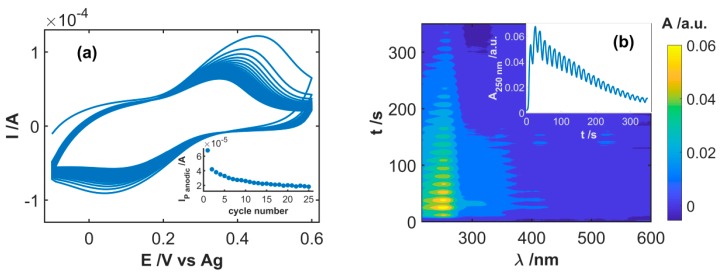
(**a**) Twenty-five successive CV scans recorded during the stabilization step of SWCNT_CFA_ in 0.1 M phosphate buffer solution (PBS). (**b**) Contour plot reporting the evolution of absorbance spectra during the whole experiment. Inset of (**a**) shows the evolution of the current intensity of the anodic peak during the 25 scans. Inset of (**b**) shows the evolution of absorbance at 250 nm recorded during the different scans.

**Figure 4 sensors-19-00518-f004:**
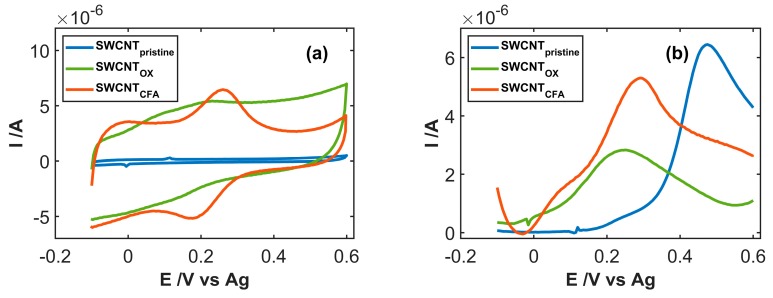
(**a**) CV scan of SWCNT-based electrodes recorded in 0.1 M PBS; (**b**) forward CV scan on the same electrodes in 2·10^−4^ M NADH, 0.1 M PBS solution, subtracted for the relevant background signal.

**Figure 5 sensors-19-00518-f005:**
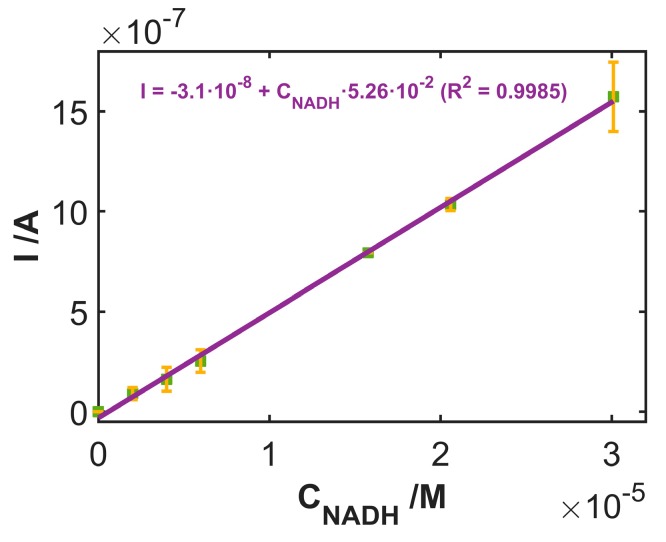
Calibration curve of the current intensity values from amperometric measurements polarizing the electrode at +0.30 V vs. the NADH concentration. Each point has been replicated three times using the same electrode for all measurements. Error bars account for standard deviation.

**Figure 6 sensors-19-00518-f006:**
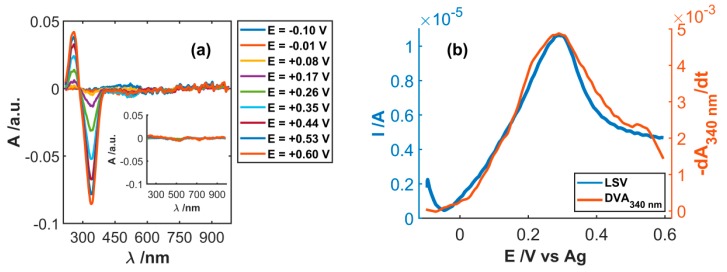
(**a**) Spectra registered during the oxidation of 3·10^−4^ M NADH and 0.1 M PBS. (**b**) Comparison of the linear sweep voltabsorptograms (LSVA) and the derivative voltabsorptogram at 340 nm. Inset in (**a**) shows the spectra registered in 0.1 M PBS.

**Figure 7 sensors-19-00518-f007:**
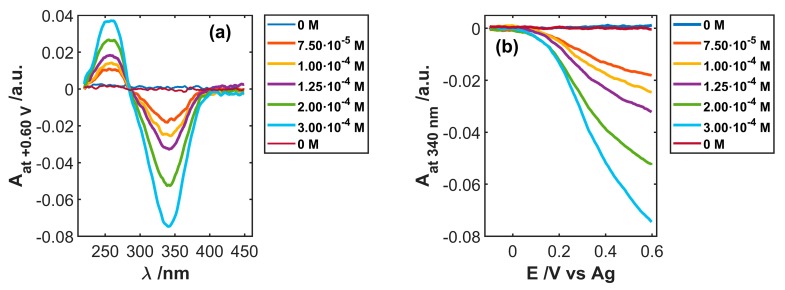
(**a**) Spectra at +0.60 V and (**b**) voltabsorptograms at 340 nm registered during the oxidation of different concentrations of NADH in 0.1 M PBS.

**Figure 8 sensors-19-00518-f008:**
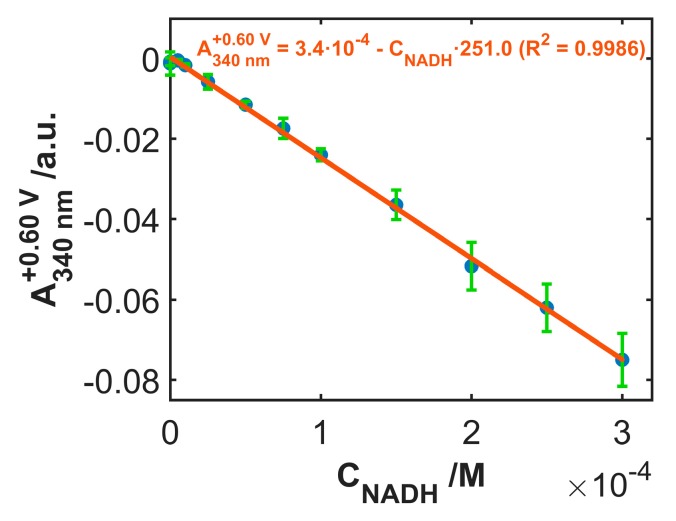
Calibration curve of the absorbance values at +0.60 V from the voltabsorptograms at 340 nm vs. NADH concentration. Each point has been replicated three times using the same electrode. Error bars, accounting for the standard deviation, are shown.

**Table 1 sensors-19-00518-t001:** Figures of merit of the three linear absorptometric calibration plots obtained by the ordinary least square regression model from data obtained using three different SWCNT_CFA_ electrodes.

	SWCNT_CFA_-1	SWCNT_CFA_-2	SWCNT_CFA_-3
**Slope /M^−1^**	−267.7	−259.6	−274.7
**Intercept /a.u.**	0.0015	0.0011	0.0007
**R^2^**	0.9998	0.9986	0.9960
**S_y/x_**	0.4·10^−3^	1.1·10^−3^	2.0·10^−3^

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
