# Peer review of "A Flexible Platform of Electrochemically Functionalized Carbon Nanotubes for NADH Sensors"

_sensors, 2019, doi:10.3390/s19030518_

Round 1
Reviewer 1 Report
Authors have presented a caffeic acid functionalized carbon nanotube system for detection of NADH.
I have few pressing concerns on the novelty of this article. There are several carbon nanotube systems for detecting different chemical moieties; the extent of literature on NADH detection is also vast. How this article differ from ref 32 of this article [Zare and Golabi, 2000] and others such as Blandon-Naranio et al [2018, doi.org/10.1155/2018/6525919].
Apart from this, there are few other comments:
Adding an illustration depicting various steps for preparing the electrochemical sensor and a cartoon with how CFA would catalyze the NADH detection and quantification would help readers immensely
Figure 5, what is the unit of concentration (uM or ug/mL)
Spectra in Fig. 7 was obtained at mM concentrations while in Fig 8 the Concentration is expressed as 10^(-4) again without any units
If I understand it correctly, this approach could determine NADH in 5-100 uM range then Authors should describe how it corresponds to the different health status of human body, viz. healthy breast cells have 99 uM NADH while cancer cells would have 163 uM
In Fig 5 Authors employed +0.3 V scanning while in Fig 5 +0.6V. CFA doesn't behave similar at the activation potentials used.
Author Response
We really appreciate Reviewer’s help that has given us the opportunity of improving the level of the manuscript. We sincerely thank the Reviewer for the insightful comments.
Authors have presented a caffeic acid functionalized carbon nanotube system for detection of NADH.
I have few pressing concerns on the novelty of this article. There are several carbon nanotube systems for detecting different chemical moieties; the extent of literature on NADH detection is also vast. How this article differ from ref 32 of this article [Zare and Golabi, 2000] and others such as Blandon-Naranio et al [2018, doi.org/10.1155/2018/ 6525919].
Answer: The SWCNT-modified electrode proposed in this paper finally consists of a flexible device for NADH detection. As it was previously described by the authors (RSC Adv. 2016, 6, 31431) the use of SWCNTs is required for the fabrication of this kind of devices. This nanomaterial is directly fixed on a non-conducting support by a high reproducible methodology, avoiding the casting of the SWCNT dispersion on that support. Drop casting, in fact, often leads to irreproducible results. This last aspect has been stressed in the revised version of the paper, in order to emphasize the reason of the use of SWCNTs for this application. Due to the effectiveness of the approach reported in ref. 32, consisting of the modification of glassy carbon electrodes by caffeic acid molecules, we decided to adapt this method to the functionalization of SWCNTs in order to improve sensitivity and reproducibility of the sensor response. In this respect we also have to stress that SWCNTs functionalized by CFA molecules allow achieving quite repeatable responses. Conversely, low repeatability was observed by bare and oxidized CNT in the paper cited by the reviewer (Blandon-Naranio et al [2018, doi.org/10.1155/2018/ 6525919]). We can suppose that the presence of organic residues most probably reduces fouling phenomena affecting the repeatability at bare and at oxidized CNT.
Apart from this, there are few other comments:
Adding an illustration depicting various steps for preparing the electrochemical sensor and a cartoon with how CFA would catalyze the NADH detection and quantification would help readers immensely.
Answer: This picture has been added as Figure S1 in Supporting Information of the revised version of the paper. As a consequence, further figures of this Section have been renumbered.
Figure 5, what is the unit of concentration (uM or ug/mL)
Answer: The referee is right, we forgot to insert the units in the calibration plot. Figure 5 was modified accordingly.
Spectra in Fig. 7 was obtained at mM concentrations while in Fig 8 the Concentration is expressed as 10^(-4) again without any units.
Answer: Figure 8 was modified to insert units; concentration values in Figure 7 were modified in order to level out the expression of the concentration in all figures to M values.
If I understand it correctly, this approach could determine NADH in 5-100 uM range then Authors should describe how it corresponds to the different health status of human body, viz. healthy breast cells have 99 uM NADH while cancer cells would have 163 uM.
Answer: To the best of our knowledge, the main application of sensors for NADH is in the development of biosensors for chemical species constituting the substrate of NADH-dependent enzymes: the enzyme, principally belonging to the dehydrogenase family, catalyze the oxidation of the substrate (i.e. analyte) due to NAD+ reduction to NADH. In this frame, the concentration of NADH produced by the enzymatic reaction strongly depends on many factors, among whom the enzymatic kinetics. But the reviewer is right when suggesting that also a direct analysis of NADH cellular levels is of interest to identify and monitor an altered status of health. Since this effect is not specifically related to breast cancer, we decided to add a brief discussion concerning this application of NADH sensors in the Introduction section, without referring to defined concentration ranges. Ref. 3 was inserted to support this approach and, as a consequence, the following references have been re-numbered.
In Fig 5 Authors employed +0.3 V scanning while in Fig 5 +0.6V. CFA doesn't behave similar at the activation potentials used.
Answer: The mechanism of charge transfer between CFA molecules and NADH, schematically reported in Figure S1 of the Supporting Information, is similar in both cases. The choice of different potentials in the two cases is due to the intrinsic nature of the two techniques, chronoamperometry and voltabsorptometry, respectively. In the former case, there is no real advantage in the application of potential values much higher than the current peak, although the limiting current has not been reached yet: the sensitivity does not increase significantly and, on the other side, species present in solution can interfere. As also reported in the paper, amperometric techniques are blind in this respect.
However, in spectroelectrochemical experiments using potential sweep techniques, the sensibility is increased at potentials higher than the current peak and the potential selectivity ascribed by suitable choice of the wavelength allows minimising the effect of interfering species. As observed in Fig. 7B, in fact, the absorbance values still increase from +0.30 to +0.60 V. For this reason, this vertex potential (+0.60 V) was selected by us to extract the absorbance values at the characteristic wavelengths of the target molecule to carry out its quantification.
Reviewer 2 Report
A brief summary
In the paper, the authors describe a feasible way to modify the SWCNT into SWCNTCFA, which demonstrates its effectiveness in sensing of NADH quantitively from both electrochemical and spectroelectrochemical studies.
Broad comments
The research is novel, well presented, and the sensing studies have been completed quantitively, though there are couple of questions need more clear clarifications from authors.
Specific comments
1. Figure 2a shows semi-reversible oxidation/reduction peaks in 0.2- 0.8V during the activation step of the SWCNT electrode. Does this suggest some reactions of the single wall carbon nanotube structures, for example, the “breakdown of the graphitic aromatic structure and formation of oxidised residues, namely aldehydic, ketonic or carboxylic groups, on SWCNTs” as described in line 191 – line 192? The authors need to assign those current peaks well.
2. In the paper, the authors claim after 25 successive CV scans in stabilization step, the CV trace get more stabilized and the variations of the current were caused by removal of weakly linked CFA to electrode surface. In other words, the gradually stabilized CV trace from last scans were attributed to CFA strongly linked to electrode surface. Considering the CV trace of CFA (Figure S2) is highly similar as activated SWCNTOX electrode (Figure 2a), and the absorbance at 250 nm decreases significantly (very close to baseline after 25 scans), there is an argument that all the CFA molecules linked on the electrode surface might be removed during stabilization process, and the finally stabilized electric CV trace/ reversible oxidation peak was contributed from bare SWCNTOX electrode. The authors may need to supply solid evidences to confirm the effectiveness of the SWCNT functionalization process.
3. As pointed out above, the arising of a reversible oxidation peak might come from SWCNTOX electrode, instead of o-quinone/o-hydroquinone redox couple from CFA linked to the electrode surface. So, the control study of SWCNTOX is necessary in Figure 4a and Figure 4b to confirm the successful functionalization of SWCNT surface by CFA molecules.
4. Incrementally enhanced spectroscopic variations were observed when concentration of NADH solution increases from 0 to 0.075 mM, while 0.100 mM seems an outlier? (Figure 7a & Figure 7b)
Author Response
We really appreciate Reviewer’s help that has given us the opportunity of improving the level of the manuscript. We sincerely thank the Reviewer for the insightful comments.
In the paper, the authors describe a feasible way to modify the SWCNT into SWCNTCFA, which demonstrates its effectiveness in sensing of NADH quantitively from both electrochemical and spectroelectrochemical studies.
Broad comments
The research is novel, well presented, and the sensing studies have been completed quantitively, though there are couple of questions need more clear clarifications from authors.
Specific comments
1. Figure 2a shows semi-reversible oxidation/reduction peaks in 0.2-0.8V during the activation step of the SWCNT electrode. Does this suggest some reactions of the single wall carbon nanotube structures, for example, the “breakdown of the graphitic aromatic structure and formation of oxidised residues, namely aldehydic, ketonic or carboxylic groups, on SWCNTs” as described in line 191 – line 192? The authors need to assign those current peaks well.
Answer: The reviewer’s hypothesis is correct: electrochemical oxidation of SWCNT leads to breakdown of the graphitic aromatic structure and formation or redox active residues, as also reported in the cited ref. 29. They are clearly evident in the voltammetric trace collected in the pure electrolyte solution, now added to Fig. 4A. A discussion concerning this figure is now added to sections 3.1 and 3.2.
2. In the paper, the authors claim after 25 successive CV scans in stabilization step, the CV trace get more stabilized and the variations of the current were caused by removal of weakly linked CFA to electrode surface. In other words, the gradually stabilized CV trace from last scans were attributed to CFA strongly linked to electrode surface. Considering the CV trace of CFA (Figure S2) is highly similar as activated SWCNTOX electrode (Figure 2a), and the absorbance at 250 nm decreases significantly (very close to baseline after 25 scans), there is an argument that all the CFA molecules linked on the electrode surface might be removed during stabilization process, and the finally stabilized electric CV trace/ reversible oxidation peak was contributed from bare SWCNTOX electrode. The authors may need to supply solid evidences to confirm the effectiveness of the SWCNT functionalization process.
Answer: The doubt of the reviewer is correct. For this reason we added to Fig. 4A the voltammetric trace collected in the pure electrolyte solution with SWCNTox. Although the overall current due to catechol residues is only partially enhanced by the functionalisation by CFA, the increase is more consistent when considering that the peak obtained for SWCNTox is actually the convolution of different contributions. For this reason we can confirm the actual functionalization of SWCNT.
3. As pointed out above, the arising of a reversible oxidation peak might come from SWCNTOX electrode, instead of o-quinone/o-hydroquinone redox couple from CFA linked to the electrode surface. So, the control study of SWCNTOX is necessary in Figure 4a and Figure 4b to confirm the successful functionalization of SWCNT surface by CFA molecules.
Answer: Voltammograms in PBS solution and in NADH/PBS solution, obtained at SWCNTOX electrode, were added in Figs. 4A and 4B as suggested by the reviewer. As well evident, both signals meaningfully change after the modification of SWCNT with CFA molecules, indicating that the modification step has been successful and the response is related to this CFA molecules anchored to the electrode surface.
4. Incrementally enhanced spectroscopic variations were observed when concentration of NADH solution increases from 0 to 0.075 mM, while 0.100 mM seems an outlier? (Figure 7a & Figure 7b).
Answer: The reviewer is right. We changed the graphs reported in Figs. 7A and 7B during the preparation of the paper, without updating the relative legends. This mistake has been corrected in this revised version.
Round 2
Reviewer 2 Report
The revisions are acceptable, and I recommend publishing this paper.